# Windsock is Dancing: Adaptive Multimodal Retrieval-Augmented Generation

**Shu Zhao**[1], **Tianyi Shen**[1], **Nilesh Ahuja**[2], **Omesh Tickoo**[2], **Vijaykrishnan Narayanan**[1]

[1]The Pennsylvania State University [2]Intel

{smz5505, vijaykrishnan.narayanan}@psu.edu

## Abstract

Multimodal Retrieval-Augmented Generation (MRAG) has emerged as a promising method to generate factual and up-to-date responses of Multimodal Large Language Models (MLLMs) by incorporating non-parametric knowledge from external knowledge bases. However, existing MRAG approaches suffer from static retrieval strategies, inflexible modality selection, and suboptimal utilization of retrieved information, leading to three critical challenges: determining **when** to retrieve, **what** modality to incorporate, and **how** to utilize retrieved information effectively. To address these challenges, we introduce `Windsock`, a query-dependent module making decisions on retrieval necessity and modality selection, effectively reducing computational overhead and improving response quality. Additionally, we propose `Dynamic Noise-Resistance (DANCE)` Instruction Tuning, an adaptive training strategy that enhances MLLMs' ability to utilize retrieved information while maintaining robustness against noise. Moreover, we adopt a self-assessment approach leveraging knowledge within MLLMs to convert question-answering datasets to MRAG training datasets. Extensive experiments demonstrate that our proposed method significantly improves the generation quality by 17.07% while reducing 8.95% retrieval times.

## 1 Introduction

While Large Language Models (LLMs) have demonstrated remarkable capabilities [1, 2, 3], they often struggle with factual accuracy [4, 5], leading to the development of Retrieval-Augmented Generation (RAG) as a solution by incorporating external knowledge into the generation process [6]. In a typical RAG pipeline, given a query, the system first retrieves relevant documents from an external knowledge base and then concatenates them with the query as context, enabling the model to generate responses grounded in the retrieved information.

While RAG has shown significant success with text-based knowledge [7, 8], real-world tasks inherently require multimodal understanding, like visual analysis or spatial reasoning [9]. This fundamental need for multimodal comprehension has led to the development of multimodal RAG (MRAG) systems that can seamlessly integrate both textual and visual knowledge in their reasoning process [10].

However, existing MRAG methods suffer from three significant limitations. First, they adopt a retrieve-for-all strategy that indiscriminately retrieves information for every query and cannot determine **when** to retrieve, even when the model's parametric knowledge is sufficient to generate accurate responses [11, 12]. It leads to unnecessary computational overhead and may introduce noise from unreliable retrievers that degrade response quality. Second, current approaches either consistently retrieve images [11, 12] or exclusively access text-based knowledge bases like Wikipedia [13, 14] without considering the specific informational needs of each query. This inflexible approach fails to recognize that different queries may require different types of information for optimal response generation, underscoring the necessity for an adaptive retrieval mechanism to make query-dependent

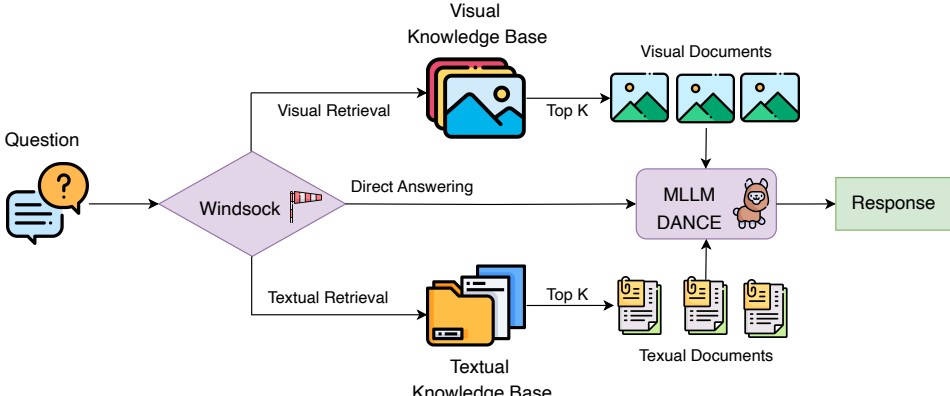

Figure 1: Framework overview of our proposed method. Given a query, Windsock adaptively selects between direct answering (no retrieval) or retrieving from either visual or textual knowledge bases, followed by MLLM instruction tuned by DANCE, generating the final response based on the query and retrieved documents.

decisions about **what** modality to select. Third, Multimodal Large Language Models (MLLMs) [15, 16] may struggle to know **how** to utilize retrieved information effectively and are sensitive to irrelevant documents [11, 17]. Either statistic-based [18] or vector-based [19, 20] retrievers may wrongly understand the query and return irrelevant documents, leading to reduce the reliability of responses and the risk of factual errors and hallucinations [21, 22].

To address these limitations, we propose Windsock, a query-aware decision module that dynamically orchestrates the retrieval process for the when and what challenges. Specifically, Windsock decides if a retrieval operation is necessary and what modality should be retrieved based on user queries. For example, queries about historical events are more likely to benefit from textual documents, while visual documents contribute to queries regarding artist's painting styles. This adaptive approach reduces computational overhead and improves response accuracy by ensuring retrieved information aligns with query intentions. Furthermore, Windsock's modular design allows it to be seamlessly integrated into open-source and proprietary MLLMs. To address the how challenge, we devise Dynamic Noise-Resistance (DANCE), an adaptive training strategy to dynamically select a modality during instruction tuning in response to retrieval noises, further enhancing MLLMs to improve their utilization of retrieved information while improving their robustness against irrelevant information.

Furthermore, we propose an automatic self-assessment approach to convert any question-answering dataset to support RAG training. Specifically, we assess the inherent ability of MLLMs to identify their incomplete knowledge by providing or withholding retrievals with different modalities. Then, we can evaluate responses to identify the best retrieval strategy and the challenging samples for training.

To summarize our contributions, we propose Windsock, a lightweight module that makes query-dependent decisions on retrieval necessity and modality selection, effectively reducing computational overhead while improving response quality. We develop DANCE, an adaptive training strategy that enhances MLLMs' ability to utilize retrievals while maintaining robustness against noise. Additionally, we introduce a self-assessment pipeline that automatically converts question-answering datasets for RAG training without relying on proprietary models, enabling efficient identification of optimal retrieval strategies and instruction tuning dataset construction. Comprehensive experiments demonstrate that our proposed method significantly improves both efficiency and response quality.

## 2 Related Work

**Multimodal Retrieval-Augmented Generation**  Multimodal Retrieval-Augmented Generation (MRAG) has emerged as a promising approach to enhance multimodal large language models (MLLMs) by incorporating external knowledge sources. Early works focused on optimizing retrieval mechanisms through end-to-end training. RA-CM3 [6] and MuRAG [23] pioneered simultaneous optimization of retrievers and generators, while RagVL [12] introduced reranking through MLLM

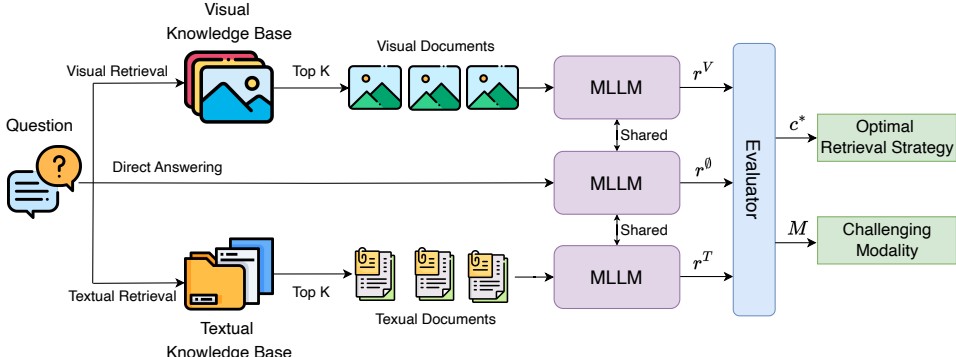

Figure 2: Data construction pipeline. An MLLM generates responses using different strategies (direct answering, visual retrieval, and textual retrieval), which are then evaluated to determine both the optimal retrieval strategy and challenging modality for training.

fine-tuning. Subsequently, WikiLLaVA [24] and EchoSight [25] advanced entity-centric retrieval by identifying and leveraging relevant contextual information. SURf [11] introduced selective information utilization to improve knowledge integration and enhance noise rejection. RIR [26] innovated by incorporating real-time information from online search engines. VisRAG [27] and VisDoM [28] specialized in multimodal document retrieval, and Re-ViLM [29] enhanced image captioning by integrating external knowledge into transformer cross-attention layers. Domain-specific applications have demonstrated MRAG's practical value, with systems like FactMM-RAG [30] improving medical radiology report generation. Recent benchmarks, including MRAG-Bench [17] and SnapNTell [31], have highlighted both the potential and limitations of MRAG systems, particularly in handling entity-centric queries and utilizing retrieved visual knowledge effectively. Most approaches employ a retrieval-for-all strategy, leading to computational overhead and potential noise injection. While recent works like ReflectiVA [14] and mR$^2$AG [13] have introduced adaptive retrieval and reranking through special tokens, they rely on expensive human or GPT-4 annotations and overlook the importance of modality selection based on query requirements. Our approach addresses these limitations by query-dependent decisions for both retrieval necessity and modality selection, thereby optimizing computational efficiency and response quality.

**Multimodal Large Language Models** Recent advances in Multimodal Large Language Models (MLLMs) have enabled systems to process and generate both visual and textual information seamlessly. Leading commercial models such as GPT-4 [15] and Gemini [2] have demonstrated remarkable capabilities across various tasks. LLaVA [16] and PaLM-E [32] have further advanced the field by introducing efficient training methods that effectively leverage pre-trained language models and visual encoders. Particularly noteworthy is the emergence of open-source alternatives like LLaMA-Vision [3], InternVL [33], and QwenVL [34], opening access to multimodal capabilities while achieving competitive performance on standard benchmarks. In this work, we focus on enhancing MLLMs' capabilities for MRAG tasks.

## 3 Method

### 3.1 Task Definition

Given an input query $Q$ consisting of text $T$ and/or image $I$, Multimodal Retrieval-Augmented Generation aims to generate a response $r$ by leveraging an external knowledge base $\mathbb{D}$, a retriever $\mathcal{R}$, and a multimodal large language model $\mathcal{G}$. The knowledge base $\mathbb{D} = \{d_1, d_2, \cdots d_n\}$ contains a collection of documents. RAG involves two key steps. (1) Retrieval: $\mathcal{R}(Q, \mathbb{D}) \rightarrow \mathbb{D}'$, where $\mathbb{D}' = \{d_1, d_2, \cdots, d_k\}$ that identifies the $k$ most relevant documents from $\mathbb{D}$ given the query $Q$, and (2) Generation: $\mathcal{G}(Q, \mathbb{D}') \rightarrow r$ that produces the final response $r$ by conditioning on both the original query $Q$ and the retrieved documents. The objective is to maximize the quality and factual accuracy of the generated response $r$ while maintaining coherence with both the input query and the retrieved knowledge. Unlike previous works employing a static retrieval strategy [24, 25, 11], our

work groups each modality into different knowledge bases and use a query-dependent module to trigger the retrieval and select the modality dynamically. To simplify but without loss of generality, we consider visual knowledge base $\mathbb{D}^I$ containing images (and/or their metadata) and textual knowledge base $\mathbb{D}^T$ including text documents.

## 3.2 Overview

As illustrated in Figure 1, given a query $Q$, the Windsock module decides if the retrieval function $\mathcal{R}(Q, \mathbb{D})$ is called. If $Q$ can be answered without external knowledge, we directly use $\mathcal{G}$ to generate a response to reduce the retrieval latency and context length. Otherwise, Windsock decides the retrieved modality type based on the query, and the retriever returns the documents from the knowledge base of the corresponding modality. Then, a response is generated based on the query and retrieved information. Moreover, we use Dynamic Noise-Resistance (DANCE) Instruction Tuning to enhance the ability to utilize retrieved documents.

## 3.3 Windsock as an Adaptive Information Retriever

To address the limitations of static retrieval strategies in multimodal RAG systems, we propose Windsock, an adaptive retrieval mechanism that optimizes both retrieval necessity and modality selection. Unlike conventional approaches that indiscriminately retrieve information for every query [11], Windsock determines when external knowledge is required and what type of information would be most beneficial for response generation. Specifically, Windsock implements a three-way classification that maps input queries to optimal retrieval strategies. Given a user query $Q$, Windsock $\mathcal{W}$ generates a retrieval type $c$ through the mapping:

$$c = \mathcal{W}(Q) \in \{\text{NA}, \text{Visual}, \text{Textual}\}. \tag{1}$$

Based on the retrieval type, a response is generated by:

$$\begin{cases} r^{\varnothing} = \mathcal{G}(Q, \varnothing), & \text{if } c = \text{NA}, \\ r^V = \mathcal{G}(Q, \mathcal{R}(Q, \mathbb{D}^I)), & \text{if } c = \text{Visual}, \\ r^T = \mathcal{G}(Q, \mathcal{R}(Q, \mathbb{D}^T)), & \text{if } c = \text{Textual}. \end{cases} \tag{2}$$

When the retrieval type is "NA," the query can be adequately answered using the MLLM's parametric knowledge, and Windsock bypasses retrieval entirely; when the retrieval type is "Visual," the query requires visual context and Windsock activates visual document retrieval to obtain the context; when the retrieval type is "Textual," Windsock triggers text-based retrieval to fetch text documents. The hybrid retrieval and other new modalities can be supported by adding new retrieval types in Equation (2). More discussions can be found in Section G.

**Discussion** This adaptive approach offers several advantages over traditional static retrieval methods. First, it reduces computational overhead by avoiding unnecessary retrievals when the MLLM's parametric knowledge suffices. Second, bypassing retrieval for queries that can be answered directly prevents potential degradation of response quality due to noisy or irrelevant retrievals [35]. Third, while transformers process both visual and textual inputs as tokens, different types of information are inherently better represented in their native modalities due to the way knowledge is structured in datasets. For example, spatial relationships might be more efficiently captured in visual form, while historical facts may be better preserved in text. Our analysis reveals that downstream datasets exhibit such implicit biases in their information requirements, which Windsock effectively captures to optimize retrieval strategy.

## 3.4 Training Dataset Curation

A critical challenge is obtaining high-quality training data that indicates the optimal retrieval strategy for each query to train the Windsock. Rather than relying on expensive annotations from commercial models [15, 2], we propose a self-assessment approach that leverages the inherent capabilities of MLLMs to automatically determine the most effective retrieval strategy, as shown in Figure 2.

For each question-answer pair $\{Q, A\}$ in a given question-answering dataset, where $Q$ and $A$ are question and answer, respectively, we generate three distinct responses by employing different

retrieval strategies, following Equation (2). To evaluate the effectiveness of each strategy, we compute quality scores for these responses by comparing them against the ground truth answer:

$$s^{\varnothing} = \epsilon(r^{\varnothing}, A)$$
$$s^{I} = \epsilon(r^{I}, A) \qquad (3)$$
$$s^{T} = \epsilon(r^{T}, A),$$

where $\epsilon$ represents an evaluation function specific to the downstream task requirements, e.g., F1 score.

Based on these scores, we determine the optimal retrieval strategy $c^*$ for each query:

$$c^* = \arg\max_{c}(s^{\varnothing}, s^{I}, s^{T}). \qquad (4)$$

The training set of Windsock is constructed as $\{Q, c^*\}$.

**Discussion** The key advantage of this self-assessment method is that it directly optimizes task performance while accounting for the specific strengths and limitations of the underlying MLLM. By comparing performance across different retrieval strategies, we can identify cases where retrieval might be detrimental (when $s^{\varnothing} > \max(s^{I}, s^{T})$) or where specific modalities are particularly beneficial (when either $s^{I}$ or $s^{T}$ outperforms others).

### 3.5 Dynamic Noise-Resistance Instruction Tuning

While Windsock effectively determines when and what information to retrieve, MLLMs still face challenges in utilizing retrieved information effectively, particularly when dealing with irrelevant content from unreliable retrievers or inappropriate chunk sizes. Traditional instruction tuning approaches are inadequate for preparing models to handle the varying quality of retrieved information [36]. To address this limitation, we propose Dynamic Noise-Resistance (DANCE) Instruction Tuning, an adaptive training strategy that enhances MLLMs' ability to ignore misleading information.

DANCE evaluates responses with different types of retrieved knowledge (visual vs. textual) and dynamically selects the modality where the MLLM struggles most:

$$\arg\min_{M}(s^{I}, s^{T}) \in \{I, T\}, \qquad (5)$$

where $M$ denotes the retrieval modality of the challenging modality. The documents corresponding to the selected modality are probably irrelevant. Note that if $s^{I}$ is equal to $s^{T}$, we randomly select one as the challenging modality. Since our primary goal is to improve the robustness of MRAG, we specifically focus on utilizing $s^{I}$ and $s^{T}$ and ignore the $s^{\varnothing}$ in the training dataset construction process. These identified challenging cases are then collected to create an instruction tuning dataset in the format $\{Q, \mathcal{R}(Q, \mathbb{D}^M), A\}$. Subsequently, we employ the standard instruction tuning pipeline to fine-tune the MLLMs, following the approach established in previous work [16].

**Discussion** The injection of noise during instruction tuning has been shown to improve model performance [37]. While recent works have proposed noise injection for robust RAG [12, 22, 36], they rely on ground truth documents provided by dataset annotators or proprietary LLM-as-a-Judge [38]. In contrast, our DANCE approach enables noise-robust training by leveraging self-assessment modality selection. Unlike SURf [11], which uses image similarity for hard example mining and requires sequential document-by-document response generation, our method efficiently identifies challenging cases through downstream tasks' evaluation metrics and processes different modalities in parallel, resulting in faster and more task-oriented training, as shown in the following section.

## 4 Experiments

### 4.1 Experimental Setup

**Datasets and Evaluation Metrics** We evaluate our proposed method on WebQA [39] and Multi-modalQA [40]. WebQA is a benchmark dataset for multi-modal and open-domain question answering. It comprises over $34,000$ training question-answer pairs, covering diverse domains and requiring

Table 1: Main Results on WebQA and MultimodalQA. *Single* and *Multiple* denote one-hop and multi-hop questions in WebQA. [†] denotes a different setting in RagVL paper that only uses validation set images to construct the visual knowledge base. For baselines without Windsock, we provide visual and textual retrievals and report the higher performance.

| Method | Generator | WebQA | | | MultimodalQA | |
|---|---|---|---|---|---|---|
| | | Single | Multiple | All | F1 | EM |
| *w.o. Instruction Tuning* | | | | | | |
| Zero-Shot | LLaVA-v1.5-7B | 59.79 | 35.20 | 42.12 | 29.94 | 27.20 |
| | LLaVA-v1.5-13B | 58.75 | 38.14 | 43.95 | 34.71 | 30.93 |
| | Qwen2-VL-7B | 61.76 | 37.09 | 44.04 | 29.37 | 26.38 |
| Vanilla RAG | LLaVA-v1.5-7B | 58.53 | 34.36 | 41.17 | 35.88 | 32.83 |
| | LLaVA-v1.5-13B | 59.85 | 35.92 | 42.66 | 44.23 | 40.16 |
| | Qwen2-VL-7B | 62.96 | 38.36 | 45.29 | 34.37 | 31.44 |
| Windsock (Ours) | Qwen2-VL-7B | **65.92** | **38.63** | **46.32** | **46.90** | **43.01** |
| *w. Instruction Tuning* | | | | | | |
| RagVL[†] [12] | LLaVA-v1.5-13B-RagVL | 57.06 | **76.18** | 65.56 | - | 76.96 |
| DANCE[†] (Ours) | LLaVA-v1.5-13B-DANCE | **67.02** | 73.70 | **69.98** | - | **77.39** |
| SURf [11] | Qwen2-VL-7B-SURf | 62.72 | 55.60 | 57.61 | 50.98 | 46.23 |
| DANCE (Ours) | Qwen2-VL-7B-DANCE | **66.42** | **57.45** | **59.97** | **51.62** | **47.01** |
| Windsock+DANCE (Ours) | Qwen2-VL-7B-DANCE | **70.12** | **59.32** | **62.36** | **52.72** | **48.20** |

Table 2: Effectiveness and efficiency of Windsock. Performance comparison using different retrieval strategies measured by average inference time per query and F1 score on WebQA.

| Retrieval | Time (s) ↓ | Single ↑ | Multiple ↑ | All ↑ |
|---|---|---|---|---|
| *w.o. DANCE* | | | | |
| NA | **0.46** | 61.76 | 37.09 | 44.04 |
| Visual | 0.67 | 64.88 | 36.46 | 44.47 |
| Textual | 0.79 | 52.87 | 36.70 | 41.25 |
| Windsock | 0.56 | **65.92** | **38.63** | **46.32** |
| *w. DANCE* | | | | |
| NA | **0.29** | 65.58 | 47.22 | 52.39 |
| Visual | 0.50 | 63.10 | 46.58 | 51.23 |
| Textual | 0.33 | 66.26 | 58.77 | 60.88 |
| Windsock | 0.36 | **70.12** | **59.32** | **62.36** |

systems to reason over text snippets and images to generate answers. MultimodalQA is a challenging question-answering dataset that requires joint reasoning with text, tables, and images. It consists of 29, 918 questions designed to simulate complex scenarios where information is integrated from multiple sources. As mentioned in the task definition, we construct knowledge bases using images and texts. Our framework can also be extended to the tabular knowledge base. For evaluation, we employ F1 score for WebQA, and both F1 score and Exact Match (EM) for MultimodalQA. These metrics assess the overlap between generated responses and ground truth answers by comparing key entities.

**Baseline Methods** We compare our approach to SOTA MRAG methods, including RagVL [12] and SURf [11]. Note that RagVL constructs the training set using ground truth annotated documents, which differ from our method. Additionally, instead of constructing an entire knowledge base (403, 277 in WebQA and 57, 052 in MultimodalQA), RagVL constructs a separate small knowledge base (4, 972 in WebQA and 384 in MultimodalQA) for validation, significantly reducing the challenging in retrieval. To make a fair comparison, we replace the original MLLM with our fine-tuned MLLM while keeping all other experimental settings unchanged. Discussions about ReflectiVA, enc-vqa [41] dataset, and infoseek [42] dataset can be found in Section I.

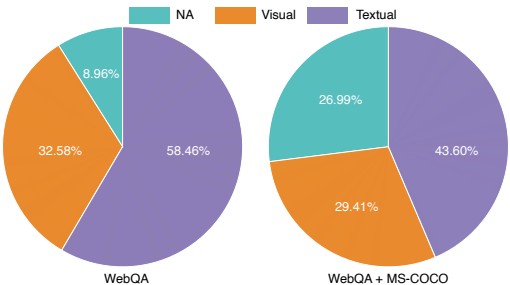

Figure 3: Windsock ratio of retrieval decisions on WebQA and WebQA+MS-COCO, showing the model's adaptive behavior across different complexities in downstream tasks.

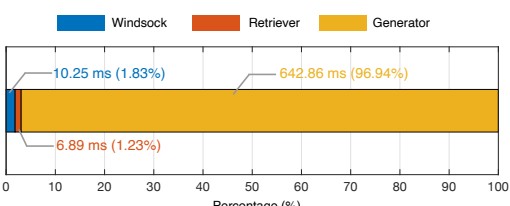

Figure 4: Pipeline runtime breakdown of Qwen2-VL, showing the proportion of total inference time consumed by each component.

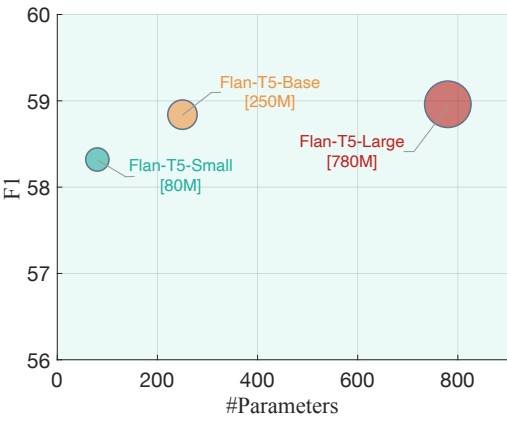

Figure 5: Performance scaling with backbone model size on WebQA, comparing F1 scores versus the number of parameters.

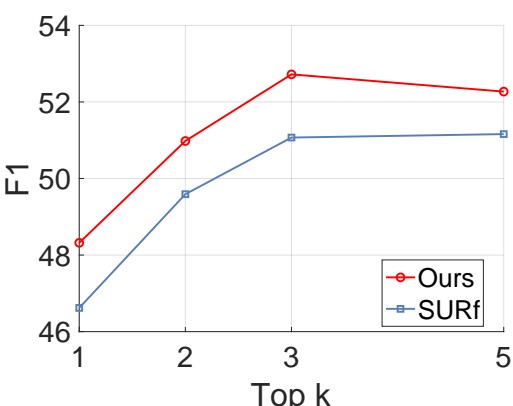

Figure 6: Impact of different top-k values on MultimodalQA.

**Implementation Details**   We use LLaVA [16] and Qwen2-VL [34] series MLLMs as the generator. The retriever is VBGE-base [19] and returns the top 3 retrievals. We use Flan-T5-Small [43] as the backbone of Windsock. To train the Windsock, we use the AdamW [44] optimizer with the learning rate $5\mathrm{e}{-}4$. The batch size is $16$, and the training epoch is $5$. We use a linear learning rate scheduler, in which the learning rate is down to $0$ at the end of training. We compute the class weights and use them in the cross-entropy loss to balance the training data [45]. For DANCE instruction tuning, we use the default LoRA [46] settings in LLaMA-Factory [47], except for setting the training epoch to $1$. If not specified, experiments are conducted on WebQA. All training experiments are conducted on $4$ NVIDIA H100 GPUs, and inference experiments use $1$ NVIDIA H100 GPU.

## 4.2   Main Results

As shown in Table 1, our method achieves substantial improvements over existing baseline approaches, demonstrating the effectiveness of our proposed method. We observe that Windsock alone provides significant performance gains compared to both zero-shot and vanilla RAG baselines, indicating that our adaptive retrieval strategy effectively reduces noise from unnecessary retrieval while maintaining high response quality. While the performance of different MLLMs varies in different downstream tasks, after instruction tuning, the overall trend in performance is upward. Most notably, combining Windsock with DANCE instruction tuning yields the best performance across all metrics. It improves $16.04\%$ and $5.19\%$ over the *w.o. Instruction Tuning* baseline on WebQA and MultimodalQA, respectively. Moreover, compared to SURf, it shows an improvement of $4.75\%$ and $1.97\%$ on WebQA and MultimodalQA, demonstrating the effectiveness of our DANCE instruction tuning method. Our proposed method can even outperform RagVL trained with ground truth documents in several metrics, showing great potential in training with noise for robust RAG systems.

Table 3: Ablation study of Windsock variants on WebQA, showing the impact of modality selection and classification strategy.

| Variant | Single | Multiple | All |
|---|---|---|---|
| Windsock | **65.92** | **38.63** | **46.32** |
| w.o. Modality Selection | 62.68 | 36.76 | 43.89 |
| w.o. One-stage | 61.31 | 37.39 | 44.13 |

Table 4: Performance comparison of different MLLMs as generators on WebQA with Windsock.

| Methods | Single | Multiple | Overall |
|---|---|---|---|
| LLaVA-v1.5-7B | 59.83 | 37.10 | 43.75 |
| LLaVA-v1.5-13B | 59.91 | 38.61 | 44.48 |
| Qwen2-VL-7B | **65.92** | **38.63** | **46.32** |

Table 5: Comparison of different instruction tuning strategies on WebQA.

| Strategies | Single | Multiple | ALL |
|---|---|---|---|
| Easy | 58.83 | 51.24 | 53.38 |
| Random | 60.98 | 53.94 | 55.12 |
| Ours | **66.42** | **57.45** | **59.97** |

Table 6: Performance comparison on the MME benchmark.

| MLLM | Perception | Cognition |
|---|---|---|
| Qwen2-VL-7B | **1423.25** | **484.00** |
| Qwen2-VL-7B-DANCE | 1394.05 | 435.71 |
| LLaVA-v1.5-13B | **1469.46** | 294.64 |
| LLaVA-v1.5-13B-DANCE | 1377.36 | **309.64** |

## 4.3 Analysis on Windsock

**Effectiveness of Windsock**   Table 2 shows the results with different retrieval types. We notice that only using a single modality knowledge base cannot consistently improve performance, and different queries may require different modality information. Moreover, our proposed Windsock outperforms them by dynamically deciding when to retrieve and what modality is useful based on the query.

**Efficiency of Windsock**   We present the average inference time per question in Table 2. While *NA* achieves the fastest inference time by omitting document retrieval, its performance is unsatisfactory. Notably, *w. DANCE* demonstrates lower inference times compared to *w.o. DANCE*. This efficiency gain stems from the shorter answer lengths in downstream tasks. WebQA typically requires phrase-level or single-sentence responses, which are more concise than the text lengths used during pre-training [3] or supervised fine-tuning stage [16]. Through instruction tuning, MLLMs learn to generate responses that align with the downstream dataset's answer length characteristics [48], typically shorter than the original MLLM outputs. Since MLLMs generate text in an autoregressive way, shorter responses require fewer token generation steps, resulting in reduced inference time. Detailed response length analysis can be found in the Appendix. Our proposed Windsock effectively balances performance and efficiency through dynamic computation allocation.

**Decisions from Windsock**   To analyze Windsock's decision-making patterns, we visualize the distribution of retrieval decisions in Figure 3. We evaluate Windsock on the WebQA validation set and an additional $2,000$ question-answering pairs sampled from the MS-COCO validation set [49]. MS-COCO samples represent simpler queries that can be answered without external knowledge, contrasting to WebQA's knowledge-intensive questions. Our analysis reveals that Windsock skips retrieval for $8.96\%$ of WebQA queries. When including simpler MS-COCO queries, the skip rate increases to $26.99\%$, demonstrating that Windsock effectively adapts its retrieval strategy based on query complexity and dataset characteristics.

**Overheads of Windsock**   Figure 4 illustrates the computational breakdown of each stage in the MRAG pipeline. The generator dominates the computational cost, accounting for $642.86$ ms $(96.94\%)$ of the total inference time. Windsock only adds $10.25$ ms $(1.83\%)$ overhead and it effectively reduces the overall computational burden by selectively skipping unnecessary retrievals, significantly improving inference speed, as shown in Table 2.

**Scalability of Windsock**   Figure 5 demonstrates how Windsock's performance scales with backbone model size, showing consistent performance improvements as the number of parameters increases. More experiments can be found in Section F. We use the language model, Flan-T5, as Windsock's backbone based on prior work [50], which shows that textual queries are crucial for understanding

Table 7: Performance comparison using ground truth documents on WebQA validation set.

| Method | F1 |
|---|---|
| Qwen2-VL-7B | 50.10 |
| Qwen2-VL-7B-DANCE | **67.35** |
| LLaVA-v1.5-13B | 44.68 |
| LLaVA-v1.5-13B-DANCE | **64.07** |

Table 8: Pipeline efficiency comparison measuring dataset construction time (in GPU hours).

| Method | GPU Hours ↓ |
|---|---|
| SURf | 32.45 |
| Ours | **15.43** |

user intent and question complexity. Nevertheless, the Windsock framework can be extended to multimodal backbones in future work.

**Variants of Windsock** To further verify the effectiveness of Windsock, we compare it to several variants, as illustrated in Table 3. *w.o. Modality Selection* denotes removing the dynamic modality selection in Windsock and keeping the ability to decide when to retrieve. The performance drop demonstrates the importance of modality-specific retrieval. *w.o. One-stage* represents a two-step decision process in which we train two separate classifiers. The first is used to decide when to retrieve, and the second is employed to determine the retrieved modality. Its lower performance suggests that joint optimization in a single stage is more effective.

More discussions about the Windsock and self-assessment framework can be found in Section H

### 4.4 Analysis on DANCE

**Backbones of Generators** Table 4 shows the results of different MLLMs as the generator. The results represent that stronger MLLMs could also achieve better performance on the MRAG task. We believe that with more SOTA MLLMs released, our proposed method could also be improved.

**Analysis of Instruction Tuning Strategy** To evaluate the effectiveness of our DANCE instruction tuning approach, we compare it with several strategies. As shown in Table 5, we consider three alternative approaches. *Easy* denotes we select the modality $M$ with the higher score in Equation (5). *Random* means randomly selecting a modality for tuning. Our proposed method achieves the best performance across all metrics, demonstrating that dynamically selecting challenging modalities leads to more effective instruction tuning.

**Impact of Retrieval Size** To investigate the impact of retrieval size on model performance, we conducted experiments varying the number of retrieved documents. As shown in Figure 6, both our method and SURf show improved performance as $k$ increases, with diminishing returns after $k = 3$. Our approach achieves optimal F1 scores at $k = 3$.

**Performance on General MLLM Benchmark** To verify how DANCE instruction tuning affects the general abilities of MLLMs, we report the results on MME benchmark [51], as illustrated in Table 6. The results show a trade-off between the abilities of downstream tasks and MLLMs' general abilities. We also notice that after DANCE tuning, the cognition score of LLaVA-v1.5-13B-DANCE outperforms the original MLLM, showing there is a potential to improve the cognition ability of LLaVA models. Discussions about improving the general ability can be found in Section J

**Performance on Ground Truth Data** To isolate the effectiveness of our instruction tuning method in information utilization (independent of retrieval quality), we conducted experiments where ground truth documents are directly provided to MLLMs, as shown in Table 7.

**Efficiency of Data Creation Pipeline** Table 8 compares dataset construction time between our method and SURf, as both approaches convert question-answering datasets to RAG format without requiring explicit RAG annotations. While SURf processes documents sequentially, generating one response per document, our pipeline processes $k$ documents per modality simultaneously. This parallel processing approach significantly reduces dataset construction time.

More discussions about the ability of noise rejection can be found in Section B.

## 5 Conclusion

In this paper, we presented Windsock, an adaptive approach to multimodal retrieval-augmented generation that dynamically determines retrieval necessity and modality selection. Combined with our DANCE instruction tuning method, this approach significantly improves both computational efficiency and response quality. Through query-dependent retrieval decisions and enhanced utilization of retrieved information, our method advances the state-of-the-art in MRAG systems while reducing computational overhead.

## 6 Acknowledgments

This work was supported in part by Semiconductor Research Corporation JUMP 2.0 PRISM Center.

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

Table 9: Performance comparison using manually created noisy documents.

| Method | F1 |
| --- | --- |
| Qwen2-VL-7B-SURf | 64.07 |
| Qwen2-VL-7B-DANCE | **66.97** |

Table 10: Results on Fine-Grained Noise Benchmarks.

| Method | MDC | CMC |
| --- | --- | --- |
| SURf | 65.5 | 58.2 |
| DANCE | **68.9 (+3.4)** | **59.5 (+1.3)** |

# A  Prompts

In this section, we provide the prompts used in this paper.

---

**WebQA with Retrievals**

You are a helpful question answerer who can provide an answer given a question and relevant context.

Question: [Question]
Context: [Context]
Provide a single sentence that answers the question based on the given context.

Answer:

---

**WebQA without Retrievals**

You are a helpful question answerer who can provide an answer given a question.

Question: [Question]
Provide a single sentence that answers the question.

Answer:

---

**MultimodalQA with Retrievals**

You are a helpful question answerer who can provide an answer given a question and relevant context.

Question: [Question]
Context: [Context]
Provide a single word or phrase that answers the question based on the given context.

Answer:

---

**MultimodalQA without Retrievals**

You are a helpful question answerer who can provide an answer given a question.

Question: [Question]
Provide a single word or phrase that answers the question.

Answer:

---

# B  Noise Rejection Ability

To verify the effectiveness of the noise rejection ability of our proposed method, we manually construct a validation set consisting of ground truth documents and irrelevant documents. Specifically,

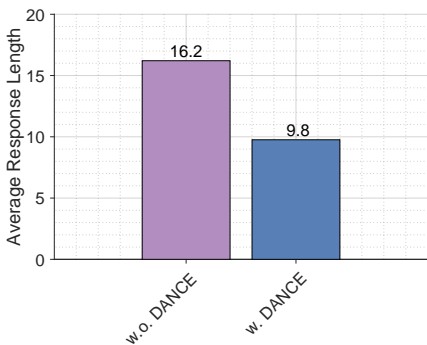

Figure 7: Comparison of average response lengths (in tokens) between models without DANCE instruction tuning (w.o. DANCE) and with DANCE instruction tuning (w. DANCE), demonstrating how instruction tuning leads to more concise responses aligned with downstream task requirements.

questions in WebQA have 1 or 2 ground truth documents. We randomly select 4 or 3 irrelevant to make sure each query has 5 documents. Then, we evaluate this dataset with SURf and our proposed method, as shown in Table 9. The results demonstrate that our method has the ability to ignore irrelevant documents in retrievals.

To create fine-grained noisy documents to verify the effectiveness, we create two fine-grained noisy benchmarks based on the MultimodalQA validation set, including cross-modal semantic conflicts (CMC) and contradictory information across multiple documents (MDC). For CMC, we prompt GPT-4o-mini to write the wrong text documents based on the ground truth captions of images. For MDC, we prompt GPT-4o-mini to identify entities or attributes in ground truth text documents and use wrong concepts to replace them. We get $521/472$ samples with MDC/CMC, and the results on EM are presented in Table 10. The results show the robustness of our method regarding complex noise.

## C    Response Length

As shown in Figure 7, we observe a significant difference in average response lengths between models with and without DANCE instruction tuning. The *w.o. DANCE* model generates longer responses with an average length of 16.2 tokens, while *w. DANCE* model produces more concise responses, averaging 9.8 tokens. This reduction in response length can be attributed to the instruction tuning process aligning the model outputs with downstream task requirements. WebQA and MultimodalQA typically expect phrase-level or single-sentence responses, which are more concise than the text lengths used during pre-training or supervised fine-tuning stages. Through DANCE instruction tuning, MLLMs learn to generate responses that better match these dataset characteristics. The shorter response lengths contribute to improved inference efficiency, as MLLMs generate text in an autoregressive manner. With fewer tokens to generate, the models require fewer generation steps, resulting in reduced inference time. This efficiency gain, combined with Windsock's adaptive retrieval strategy, helps optimize the overall computational performance of our system while maintaining response quality.

## D    Qualitative Results and Error Analysis

Figure 8 presents several examples from WebQA and MultimodalQA that illustrate both successful cases and typical failure modes of our system. Through analysis of these cases, we identify two main types of errors.

**Misdirection from Windsock**    In some cases, Windsock fails to determine the optimal retrieval strategy. For example, in the "Ben Pizza" question, Windsock does not select the correct modality. However, DANCE-tuned MLLM can ignore the irrelevant contexts and generate a correct response.

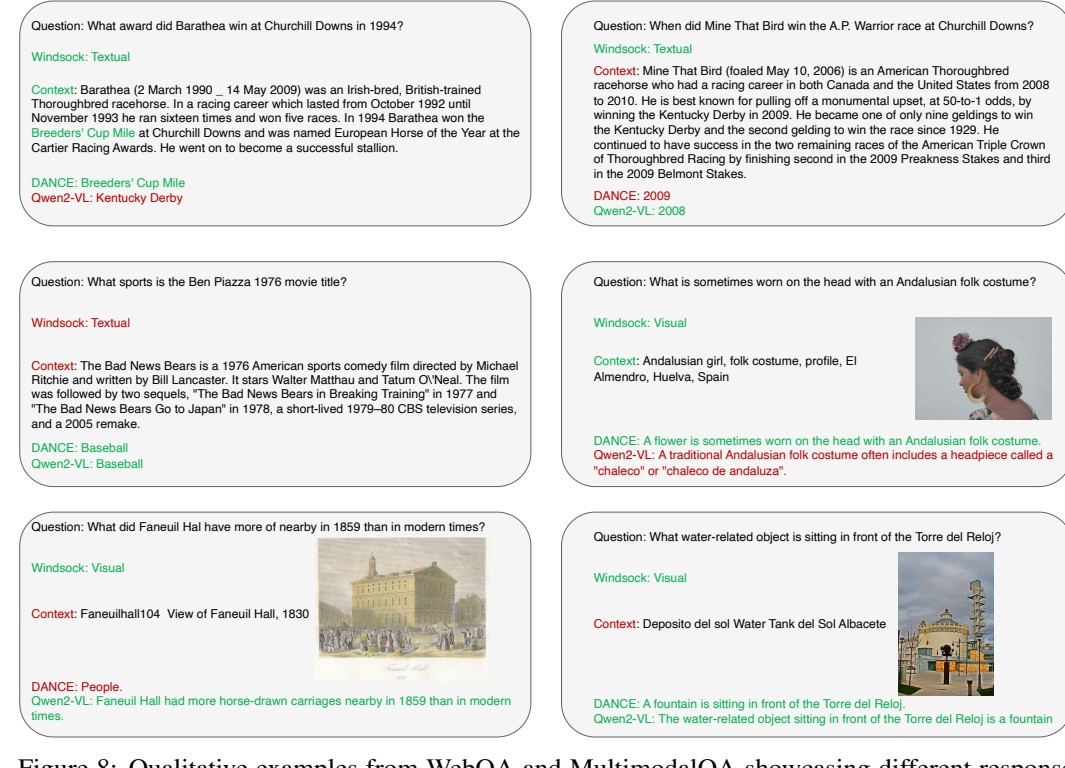

Figure 8: Qualitative examples from WebQA and MultimodalQA showcasing different response patterns between DANCE-tuned and base models. Each example includes the question, retrieved context, responses from both models, and Windsock's modality selection. Examples illustrate both successful cases and common failure modes in our system.

Table 11: Performance comparison of different retrieval models.

| Retriever | MRR@1 | MRR@5 | Recall@1 | Recall@5 | mAP@5 | NDCG@5 |
|---|---|---|---|---|---|---|
| CLIP ViT-L-14-336 | 9.72 | 13.69 | 8.18 | 17.14 | - | - |
| VisualBGE Base | 46.28 | 56.50 | 36.16 | 64.30 | 50.63 | 55.91 |
| Marvel ANCE | 55.44 | 65.72 | 43.59 | 73.95 | 59.20 | 64.67 |

**Distraction from Generator** Even when Windsock correctly identifies the retrieval strategy and relevant documents are retrieved, the generator can sometimes be misled by noisy information in the documents. For instance, in the "Faneuil Hal" question, the irrelevant retrieval misleads the DANCE-tuned MLLM.

## E  Different Retriever Performance Analysis

We evaluated three different retriever models to assess their effectiveness in our multimodal retrieval system: CLIP [20], VisualBGE [19], and Marvel [52]. The performance is measured using Mean Reciprocal Rank (MRR) and Recall metrics at two different cutoff points (top-1 and top-5). As shown in Table 11, Marvel demonstrates superior performance across all metrics, achieving the highest scores with an MRR@1 of $55.44\%$ and Recall@5 of $73.95\%$. This indicates its strong ability to rank relevant documents highly and retrieve them effectively. VisualBGE shows competitive performance as the second-best option, with an MRR@1 of $46.28\%$ and Recall@5 of $64.30\%$. In contrast, CLIP ViT-L-14-336 exhibits significantly lower performance across all metrics, suggesting it may be less suitable for this specific retrieval task. In this paper, we use VisualBGE as the default retriever, other retrievers [53, 54, 55, 56] may increase the system's performance and it can be future work.

Table 12: More results using different backbones of Windsock.

| Model | F1 |
|---|---|
| DistilBERT-Base | 55.5 |
| CLIP ViT-B/16 | 49.2 |
| Flan-T5-Small | **58.3 (+2.8)** |

Table 13: Hybrid Retrieval Results.

| Retrieval | F1 | Ratio |
|---|---|---|
| NA | 56.8 | 10.3 |
| Visual | 58.0 | 28.5 |
| Textual | 56.7 | 36.3 |
| Hybrid | 57.1 | 24.9 |
| Windsock | **59.1 (+1.1)** | - |

## F    More Results of Using Different Backbones of Windsock

Windsock is a novel query-dependent module in which the input is the query, and the outputs are the decisions on retrieval necessity (when) and modality selection (what). Compared to previous fixed and modality-agnostic methods, Windsock effectively reduces computational overhead and improves response quality. In this paper, we use Flan-T5-Small as the default model. We conduct experiments using DistilBERT and CLIP. The results presented in Table 12 indicate that Flan-T5-Small outperforms DistilBERT and CLIP since DistilBERT is a smaller model than Flan-T5, and CLIP is trained for the image/text alignment task. We believe the understanding ability of backbone models is a key point to improve Windsock's performance. A larger model could perform better, but it also introduces more inference latency.

## G    Hybrid Retrieval and More Modalities

Existing dataset construction pipelines, like WebQA [39] and MultimodalQA [40], typically instruct annotators to create questions answerable by either images or text rather than requiring both simultaneously. Encyclopedic-VQA [41] and InfoSeek [42] require models to identify entities in images and retrieve knowledge from Wikipedia to get textual descriptions. Answering questions in these datasets does not need to retrieve both visual and textual information. It is a limitation of dataset construction, not a limitation of Windsock. We notice a benchmark, MRAMG-Bench [57], requiring models to always retrieve multimodal information for multimodal generation. Using WebQA's visual retrieval, WebQA's textual retrieval, and MRAMG-Bench, we randomly sample 2,000 data points from each as the training set. Then, we randomly sample 300 data points from each as the validation set. Extending Equation (2), we train a Windsock supporting NA, Visual, Textual, and Hybrid retrieval, as shown in Table 13. These results demonstrate the scalability of Windsock while maintaining the superior inference time.

The scalability of Windsock makes it easy to expand to more modalities. We use image and text modalities because they are the most common in MLLMs. We survey the RAG datasets and do not find one that supports more than three modalities. Inspired by the video classification task, we observe that WebQA contains questions that require more than 2 images. We concatenate these images and view them as a "pseudo video" to create the third modality in WebQA. We randomly sample 2,000/300 data points from each modality as the training/val sets. To build the video indexing, we use VBGE to extract image features and add them as the video embedding. The results are presented in Table 14. These results shows that Windsock's flexibility of modalities and can be scaled up on more modalities.

Table 14: Results using More Modalities.

| Retrieval | F1 | Ratio |
|---|---|---|
| NA | 47.1 | 9.1 |
| Visual | 49.8 | 29.4 |
| Textual | 45.5 | 46.8 |
| Video | 49.4 | 14.7 |
| Windsock | **51.0 (+1.2)** | - |

Table 15: Performance Comparison of using GPT-4o and Windsock as the decision module.

| Method | EM |
|---|---|
| GPT-4o | 40.7 |
| Windsock | **43.2 (+2.5)** |

Table 16: Results on Enc-VQA using ReflectiVA and the optinal Windsock.

| Method | E-VQA | NA Ratio |
|---|---|---|
| ReflectiVA | 35.5 | 0% |
| + Windsock | **36.9 (+1.4)** | **7.6%** |

## H   Robustness of Self-Assessment

Our method improves the robustness while maintaining efficiency in two complementary ways: For cases where retrieval transforms incorrect answers to correct ones, Windsock learns to identify these patterns and trigger appropriate retrieval strategy, addressing scenarios where the MLLM's parametric knowledge is insufficient. For cases where answers remain incorrect despite retrieval since we comprehensively evaluate the response with all retrieval scenarios in Equation (3) and Equation (5), it indicates that it is too difficult to answer for the MLLM, which requires larger MLLMs or continual pre-training techniques to improve the performance of base models. We assign "NA" labels to skip retrieval to improve efficiency. Windsock ensures our system learns from successful retrieval patterns while making practical decisions for challenging cases that represent current model limitations. Moreover, this self-assessment method successfully connects the internal parametric knowledge with the downstream dataset preference through Windsock. We randomly sample $4,000/600$ training/val samples from MultimodalQA. Instead of Windsock, we prompt GPT-4o to make the retrieval decision with a question. The results are presented in Table 15.

We analyze the errors of GPT-4o and find two error patterns. First, GPT-4o is more powerful than the generators in this paper, e.g., Qwen2-VL-7B. Questions that require retrievals for Qwen2-VL may be directly answered by GPT, leading to a wrong NA label. Second, some questions can be answered by either images or texts from GPT's perspective. Due to the dataset bias, one of the modalities may not exist in downstream datasets, leading to wrong modality selection. Self-assessment and Windsock capture the bias and successfully make correct decisions regarding retrievals by connecting the internal parametric knowledge with the downstream dataset preference.

## I   Compared to ReflectiVA

ReflectiVA [14] retrieves entity textual descriptions that do not involve scenarios that require modality selection. It uses the MLLM as a point-wise reranker, first using the MLLM to decide if the retrieval is triggered and then generating a special token to filter out irrelevant documents one by one, significantly increasing the inference latency, resulting in an overall inference time of $5$ to $7$ times slower than our method ($\sim 80$ GPU hours using H100). ReflectiVA achieves nearly $0\%$ NA retrieval because it simply assigns Retrieval/NA labels to training samples in Infoseek, E-VQA, and LLAVA datasets. This fixed and modality-agnostic method cannot handle the increasing MLLM's abilities compared to our method, which connects the internal parametric knowledge with the downstream dataset preference. Notably, using ReflectiVA as the generator, Windsock is compatible with ReflectiVA, as shown in Table 16.

In the main experiments, We use WebQA and MultimodalQA since they contain samples that require image or text retrieval, which is a more challenging and realistic scenario. INFOSEEK [42] and Enc-VQA [41] require models to identify entities in images and retrieve knowledge from Wikipedia to get

Table 17: Results on the subset of Enc-VQA.

| Retrieval | Performance |
|---|---|
| NA | 17.0 |
| Textual | 19.6 |
| Windsock | **20.6 (+1.0)** |

Table 18: Mixing general instructions improves the general ability.

| MLLM | MME-P | MME-C | MMMU | MMStar |
|---|---|---|---|---|
| Qwen2-VL-7B | 1423.25 | 484.00 | 54.1 | 60.7 |
| + DANCE | 1394.05 | 435.71 | 49.7 | 56.1 |
| + DANCE + LLAVA-Instruct | 1401.39 | 457.91 | 51.3 | 57.3 |

Table 19: Statistics on the knowledge bases.

| Dataset | # Visual | # Textual |
|---|---|---|
| WebQA | 403,277 | 544,489 |
| MultimodalQA | 57,052 | 218,258 |

textual descriptions, which do not involve scenarios that require modality selection. Windsock can be used to decide when to retrieve these datasets. To verify the effectiveness and efficiency of Windsock on them, we randomly sample from Enc-VQA and obtain 2,000/200 samples as training/val sets to train the Windsock. The generator is pre-trained LLaMA-3.1-8B. The results are presented in Table 17. The results demonstrate the effectiveness and efficiency of Windsock on various datasets.

## J   General Ability Improvement

There is a performance trade-off between specific tasks and general tasks: tuning on downstream tasks may lead to decreased performance on general benchmarks, as demonstrated by recent works. To improve the RAG perf-ormance while maintaining the general ability, we mix the instructions from Equation (5) with the LLaVA-Instruct dataset and fine-tune the model. We also add two extra general benchmarks MMMU [58] and MMStar [59]. The results are shown in Table 18. This demonstrates that mixing general instruction data could improve the instruct-tuned models' general ability.

## K   Knowledge Base Construction and Windsock Decision

The visual knowledge base contains image file paths and corresponding captions. The textual knowledge base contains text chunks. We use `IndexFlatL2` from faiss library to construct the indexing. The number of data in databases is listed in Table 19.

In our implementation, we use VBGE-base as our default multimodal retriever, which supports both modalities with a unified embedding space. During the preprocessing (indexing) stage, we use VBGE to encode images from the visual database and texts from the textual database to embedding vectors and store them in the corresponding vector database. During the retrieval stage, the query text with an optional query image is encoded into the query embedding vector. If Windsock selects visual/textual retrieval, we calculate the cosine similarity between the query vector and embedding vectors from the visual/textual database and return top k results. We will add these implementation details to the revised manuscript, including pseudo code for the retrieval process.

## L   Limitations and Future Work

While our proposed method shows promising results, several limitations should be noted. First, in this work, we only focus on the vision and language modalities. Our method can extend to more modalities like audio or tabular data, which can be used in future work to verify the effectiveness and efficiency of more modalities. Second, we do not elaborate on a complex pipeline to solve the single-hop and multi-hop questions separately. Specific designs for multi-hop questions may further improve the performance. Third, Windsock currently makes binary decisions about modality selection (visual vs. textual). This may be suboptimal for queries that could benefit from varying degrees of multimodal information. Future work could explore more nuanced retrieval strategies that allow for weighted combinations of different modalities.

# M   Broader Impacts

The Windsock framework introduced in this paper has several significant positive societal and technological impacts. This approach substantially reduces computational overhead while improving response quality by intelligently determining when retrieval is necessary and which modality to select. This efficiency gain reduces energy consumption and carbon footprint, supporting more sustainable AI deployment.

The adaptive nature of Windsock makes advanced multimodal AI systems more accessible to users with limited computational resources, helping democratize access to state-of-the-art AI technologies. In educational settings, improved response accuracy with reduced latency enhances the learning experience, while in research environments, it enables more efficient information retrieval and knowledge synthesis across modalities.

Furthermore, the DANCE instruction tuning approach improves robustness against misinformation by teaching models to better distinguish between relevant and irrelevant retrieved information. This framework's modular design allows seamless integration into various existing systems, maximizing its practical impact across different applications and domains.

However, the system may inadvertently amplify existing biases in training data through its selective retrieval decisions, potentially reinforcing problematic information patterns. Additionally, as the technology becomes more widely adopted, there is a risk of increasing societal dependency on automated information filtering systems that operate as black boxes to most users.

