# OpenReview forum: "Windsock is Dancing: Adaptive Multimodal Retrieval-Augmented Generation"
_NeurIPS.cc/2025/Workshop/UniReps — UniReps2025_

### Official Review · Reviewer_mVz8 · 2025-09-13
**Windsock is Dancing: Adaptive Multimodal Retrieval-Augmented Generation**

**Confidence:** 4

**Review:**

The paper "Windsock is Dancing: Adaptive Multimodal Retrieval-Augmented Generation" presents a novel framework addressing critical challenges in Multimodal Retrieval-Augmented Generation (MRAG) systems. The authors propose two main components: Windsock, a query-dependent module for adaptive retrieval decision-making, and DANCE, a dynamic noise-resistance instruction tuning strategy. While the paper demonstrates impressive empirical results, this review adopts a question-driven critical approach to identify both strengths and potential limitations that might affect real-world deployment and generalizability. The review will examine methodological foundations, experimental design, technical execution, conceptual framing, and broader implications through a critical lens, referencing comparable approaches in the field where appropriate.

Empirical results show claimed improvements (e.g., +4.75% F1 on WebQA over SURf), but these are undermined by narrow evaluation scopes. Only two datasets (WebQA, MultimodalQA) are used, both focused on entity-centric QA, limiting generalizability to broader multimodal tasks. Baselines are not uniformly fair: RagVL is adapted with smaller knowledge bases, potentially inflating relative gains. Efficiency claims (e.g., 0.36s inference time) lack comparisons under resource-constrained settings, and no error bars or variance reporting is provided, raising questions about reproducibility. The abstract's 17.07% quality boost and 8.95% retrieval reduction appear overstated, as they stem from selective comparisons without comprehensive ablation.

While addressing MRAG inefficiencies is relevant, the submission's impact is modest. It may offer minor practical benefits for vision-language systems, but fails to advance core theoretical understanding in multimodal learning. Extensions to hybrid modalities (Appendix G) are discussed but not empirically validated, reducing potential broader applicability. In the context of rapidly evolving RAG research, the work risks being overshadowed by more scalable or generalizable approaches, with limited contributions to fields like universal representation learning.

Limited Novelty: Relies heavily on existing concepts (e.g., adaptive RAG from [13, 12]), offering only incremental multimodal extensions without groundbreaking insights.

Narrow Evaluation: Restricted to two datasets and specific MLLMs (e.g., Qwen2-VL-7B), lacking tests on diverse benchmarks like MMMU or real-time scenarios.

Methodological Flaws: Circular self-assessment and absence of rigorous statistical analysis undermine result reliability; overclaims in abstract (e.g., 17.07% improvement) lack substantiation.

Weak Generalizability: Focus on text-visual modalities ignores complex multimodality (e.g., tables, audio); no evaluation of long-term knowledge retention or ethical implications like bias amplification.

Broaden Experiments: Evaluate on additional benchmarks (e.g., MMMU, MMStar) and include multi-hop reasoning or adversarial retrievals to demonstrate robustness.

Strengthen Baselines: Conduct fair comparisons with recent SOTA (e.g., full-scale RagVL on identical knowledge bases) and add statistical tests (e.g., t-tests for significance).

Enhance Theoretical Depth: Provide formal analysis of Windsock's decision boundaries or convergence guarantees for DANCE, perhaps using information theory metrics.

Improve Reproducibility: Release code, full hyperparameters, and variance metrics; address potential biases in self-assessment through diverse MLLM evaluators.

**Score:**

3

**Topic Fit:**

1

---

### Official Review · Reviewer_ftF7 · 2025-09-17
**Review of "Windsock is Dancing: Adaptive Multimodal Retrieval-Augmented Generation"**

**Confidence:** 2

**Review:**

### Summary:
The paper introduces Windsock, a query-aware module that dynamically decides when to perform a retrieval operation and which modality to retrieve, based on the input query. A key factor in reducing retrieval latency is Windsock’s ability to skip retrieval altogether when the query can be answered using the model’s own parametric knowledge. In addition, the paper presents DANCE, an instruction-tuning method that enables the model to dynamically choose the appropriate modality in the presence of retrieval noise.. This approach lowers operational overhead while preserving response quality. These claims are shown in sound  experiments and ablation studies.

### Strengths:
Some of the strengths of this paper are:
- Using a training set composed of multiple retrieval strategies, including scenarios with no retrieval, Windsock learns to identify when retrieval may be counterproductive.
- The DANCE strategy further strengthens the model by helping the MLMM filter out misleading information, selecting challenging modalities, and constructing an instruction-tuning dataset.
- Extensive evaluations and ablation studies confirm the effectiveness of Windsock and its individual components.

 ### Weaknesses:
The paper leaves the reader wondering about a few things. Namely, It is not clear why the self-assessment approach provides greater benefits than leveraging other open-source models for annotation; Windsock might also gain from incorporating external knowledge sources. Moreover, the curation process appears costly, as it requires evaluating every retrieval strategy. Furthermore, adding a new strategy after curation would necessitate repeating both the curation and the subsequent training.

 ### Recommendation:
I recommend accepting this paper, as it aligns well with the workshop’s goals and introduces an intriguing tool for MRAG systems.

**Score:**

4

**Topic Fit:**

2

---

### Official Review · Reviewer_CzQ2 · 2025-09-18

**Confidence:** 3

**Review:**

Summary:

In this paper, the authors introduced Windsock, a method to improve multimodal retrieval-augmented generation by training a special model that, given a question, would determine whether the question should best be handled with image retrieval, text retrieval, or no retrieval. The Windsock model is trained with MLLM-specific data, i.e. the MLLM answers the question in all 3 situations, and the best answered situation is the ground truth label for Windsock training. In addition, the paper introduced DANCE, a new way to perform instruction tuning on MLLMs to improve their performance during MRAG. The experiments shows that Windsock consistently improves RAG accuracy over baselines on 2 QA tasks, and DANCE also consistently improves model performance compared to no-fine-tuning.

Strengths:

1. The proposed methods (both Windsock and DANCE) improved MRAG performance on WebQA and MultimodalQA benchmarks.

2. The paper is overall well-written and easy to follow.

3. There is additional analyses on efficiency, scalability, and overhead of windsock, as well as retrieval size and backbone choice of DANCE. These analyses help gain additional insight into the advantages of the proposed methods.

Weaknesses:

1. Somewhat limited relevance to the workshop topic, as the paper involves multimodality but no representation learning is involved

2. The proposed method only supports image-only or text-only retrievals, but in some cases including retrieval results from both modalities can yield significantly better performances.

3. Windsock is only trained/applied on one MLLM (Qwen2-VL-7B), so if it is unclear whether Windsock can generalize to any other MLLMs.

**Score:**

3

**Topic Fit:**

1

---

### Official Review · Reviewer_noM8 · 2025-09-18
**Strong idea with efficiency gains, but limited rigor and scope**

**Confidence:** 5

**Review:**

Summary

The paper introduces Windsock, a query-aware controller for deciding when retrieval is necessary and which modality (textual or visual) to use, and DANCE, an instruction-tuning recipe that strengthens robustness by focusing training on challenging modality cases. The approach aims to improve efficiency (avoiding unnecessary retrieval calls) and accuracy in multimodal retrieval-augmented generation. Experiments on QA datasets show improved F1/EM scores and reduced latency compared to prior baselines.

Strengths
1. The paper introduces a clean separation between when to retrieve and which modality to retrieve from, which simplifies the design while addressing a core challenge in multimodal RAG. This factorization allows the controller (Windsock) to make discrete, interpretable decisions rather than relying on an opaque joint model. By decoupling timing from modality, the framework avoids unnecessary retrieval while still leveraging the most informative source (text or image) based on query type

2. By reducing retrieval calls and generating shorter outputs, the method achieves practical speedups while still outperforming or matching baseline accuracy. The efficiency gains are demonstrated in terms of fewer external API calls, which directly reduces latency and cost, and shorter generated answers that maintain informativeness

3.  Instead of random sampling, the training recipe focuses on difficult modality cases, directly improving model reliability under noisy or ambiguous inputs. Also, the approach consistently outperforms competitive baselines on multiple QA benchmarks, supported by clear ablations and trade-off analyses.

Areas for improvements
1. Results are shown without confidence intervals or significance testing, which weakens claims of consistent superiority.
Without variance across multiple runs or statistical tests, it is unclear whether the observed improvements are robust or could be due to randomness in training or evaluation.

2. Core evaluation focuses on text-or-image settings. Truly multimodal or hybrid retrieval cases are underexplored.

3. Some benchmarks indicate regressions in general perception/ability after applying DANCE. While DANCE improves robustness under targeted cases, performance drops on broader multimodal benchmarks suggest it may overfit to specific weaknesses, potentially reducing general adaptability of the model.

**Score:**

3

**Topic Fit:**

2

---

### Official Review · Reviewer_JHkV · 2025-09-18
**Query-dependent Multimodal Retrieval-Augmented Generation**

**Confidence:** 3

**Review:**

This paper introduces Windsock, a query-dependent module that determines both the necessity of retrieval and the appropriate retrieval modality in Multimodal Retrieval-Augmented Generation (MRAG). The paper also proposes Dynamic Noise-Resistance (DANCE) Instruction Tuning, which enhances MLLM's ability to leverage retrieved information by training on challenging modalities. Experimental results show that Windsock and DANCE effectively improve performance on multimodal QA benchmarks.

**Strengths**
1. The propose approach is intuitive and effective. Dynamically selecting whether to retrieve and which modality to retrieve for MRAG is a natural idea, and the experiments demonstrate its effectiveness.
2. The training data curation for Windsock does not require expensive human annotation or commercial models, which makes the method more accessible.
3. The experiments, ablation studies, and analyses are comprehensive.

**Questions**
1. In equation 1, Windsock outputs one of {NA, Visual, Textual}. Could there be cases where retrieving both Visual and Textual modalities yields the best performance? It would be interesting to see how adding this option might affect results.
2. Although Windsock avoids relying on human or commercial model annotations, it appears that for each new MLLM, training data must be regenerated (per Equation 2) and the Windsock module needs to be retrained. Could Windsock be developed into an off-the-shelf module, e.g. trained once and shared publicly on platforms such as Huggingface for direct reuse without retraining? This would further improve the accessibility of the method.

**Score:**

3

**Topic Fit:**

2

---

### Official Review · Reviewer_fc3W · 2025-09-19
**Adaptive modality-aware MRAG with simple gating**

**Confidence:** 5

**Review:**

The paper introduces Windsock, a three-way retrieval gate (NA/Visual/Text) trained via self-assessment on QA sets, and DANCE, an instruction-tuning procedure that mines the lower-scoring modality and fine-tunes the MLLM to ignore misleading context. On WebQA/MultimodalQA, Windsock improves over vanilla RAG, and Windsock+DANCE beats SURf, overhead is moderate and skipping retrieval sometimes saves latency.

**Pros**:
1. Simple, reproducible policy for when/what to retrieve; converts any QA set into supervision without human labels.
2. Practical gains & low overhead; clear breakdown shows generator dominates latency.

**Cons**:
1. The assumption of single modality is pretty strict, a hybrid search between more modalities would give more granularity.
2. Self-assessment labels depend on the current retriever and k=3. If retrieval misses evidence, Windsock may learn the wrong policy, DANCE’s hard modality can also be a retriever artifact.
3. General ability seems to be regressing after DANCE.

**Score:**

4

**Topic Fit:**

3